# In Vitro Fracture Resistance of Endodontically Treated Premolar Teeth Restored with Prefabricated and Custom-Made Fibre-Reinforced Composite Posts

**DOI:** 10.3390/ma14206214

**Published:** 2021-10-19

**Authors:** Michal Bialy, Sara Targonska, Agnieszka Szust, Rafal J. Wiglusz, Maciej Dobrzynski

**Affiliations:** 1Department of Pediatric Dentistry and Preclinical Dentistry, Wroclaw Medical University, Krakowska 26, 50-425 Wroclaw, Poland; michal.bialy@umed.wroc.pl (M.B.); maciej.dobrzynski@umed.wroc.pl (M.D.); 2Institute of Low Temperature and Structure Research, Polish Academy of Sciences, Okolna 2, 50-422 Wroclaw, Poland; 3Faculty of Mechanical Engineering, Wroclaw University of Science and Technology, Wybrzeze Wyspianskiego 27, 50-370 Wroclaw, Poland; agnieszka.szust@pwr.edu.pl

**Keywords:** premolars, endodontic treated, fibre-reinforced composite post, strength test

## Abstract

(1) Background: The study aimed to compare and analyse the differences between the features of prefabricated fibre-reinforced composite (FRC) posts and custom-made FRC posts in the form of a tape and confirm the necessity of using FRC posts in teeth treated endodontically in comparison to direct reconstruction with a composite material. (2) Methods: Sixty premolars after endodontic treatment were used. The teeth were divided into four groups (n—15). Group 1: teeth with embedded prefabricated posts (Mirafit White); group 2: teeth with embedded prefabricated posts (Rebilda); group 3 teeth with embedded custom-made posts in the form of a tape (EverStick); group 4: teeth without a post restored with composite material. The compressive strength of the teeth was tested using the Instron-5944 testing machine until the sample broke. The crystal structure of the investigated posts was detected with the X-ray diffractometer (3) Results: During the experiment, the maximum values of forces at which the damage of the restored premolar teeth after endodontic treatment occurred were obtained. The best results were obtained for teeth rebuilt with Rebilda Posts (1119 N), while teeth with cemented Mirafit White posts were the weakest (968 N). Teeth without an embedded FRC post, rebuilt only with light-cured composite material, obtained the lowest value—859 N. (4) Conclusions: The use of FRC posts increases the resistance to damage of an endodontically treated tooth when compared to direct restoration with light-cured composite material.

## 1. Introduction

Teeth after endodontic treatment are structurally different from vital teeth and require specialized restorative materials [1,2]. The major difference is a consequence of the loss of dental structures caused by caries, fractures of previous restorations and endodontic procedures. Consequently, the risk of fracture increases after endodontic treatment. [3,4]. Finally, their mechanical and strength properties are directly proportional to the volume of origin tooth tissue as well as the chosen treatment technique [5,6,7,8,9,10,11].

The insertion of a post significantly increased the fracture resistance of nonvital premolars [12,13,14]. One of the main goals of posts’ presence is to ensure the dental materials’ retention while the lost dental crown is being reconstructed. A post allows for the appropriate stress distribution in the root and may be used to support single crowns and bridges. However, according to numerous researchers, a post does not strengthen the tooth structure, it merely provides the appropriate retention for the material used to restore a crown foundation [15,16,17,18]. These findings could be caused by the fact that preparation of space for the crown-root inlay may lead to the diminishing of the remaining part of the tooth structure and increase the risk of the root breaking. This emphasizes the significance of preserving the original anatomy of the root canal and minimizing dentin loss in the reconstruction following the completed endodontic treatment.

There are several factors that could influence the root canal filling related to the mechanical preparation of the post space such as the twisted or vibrated areas [19]. The filled root canals can become contaminated again by microorganisms during the preparation of the canal for the post, when the remaining apical part of the root filling is of an inappropriate density or length. It is a risk factor for the development of periapical inflammation. Santos et al. assess that root canal-filled teeth coronally restored with a rubber dam did not develop severe periapical inflammation but root canal fillings exposed to microbial challenge with exposure to saliva developed severe periapical inflammation [20]. They concluded that the current methods of root canal filling only partially inhibit the penetration of microorganisms into the crown and the development of apical periodontitis after treatment, highlighting the risk of periapical pathology in case of a break in the chain of asepsis during post preparation.

It was stated in several cases that endodontic treatment by using one fibre-reinforced composite (FRC) post may not be enough to ensure suitable mechanical friction in the case of root canals characterized by an unusual and irregular shape. To solve the problems with the unusual shape of root canals, an elastic FRC post (GC Europe, Leuven, Belgium) was introduced [21]. It should be noted, the elastic FRC post can be individually customized and their ability to bond and bend is superior to the prefabricated posts that commonly appear on the market.

The presented study is focused on investigating and comparing the crack resistance of endodontically treated premolar teeth reinforced by different available FRC posts. The null hypotheses were: (1) The application of FRC posts increases the resistance to damage of an endodontically succeeding treated tooth when compared to direct restoration with light-cured composite material and (2) no significant difference was noted in the case of using more elastic fibre posts.

## 2. Materials and Methods

### 2.1. Preparation of Materials

In the research, 60 first premolar teeth extracted for orthodontic or periodontal indications were used. The teeth neither showed any signs of caries, root cracks, resorption nor were they treated before. Each of the teeth had two roots and two canals, the buccal and the palatine one. All of the selected teeth had similar dimensions: the length of the crowns was 8.5 mm, the roots’ lengths were 14 mm and the diameter of the mesial-distal crowns was 7 mm. They were kept in 0.9% NaCl solution. The root surface was purified of soft tissue by using the hand scaler. Prior to strength tests, they were treated endodontically. The working length was determined using a direct method related to subtracting 1 mm from the actual root length set by introducing a no. 10 K-file (Maillefer-Dentsply, Ballaigues, Switzerland). Irrigation was performed after every change of instrument with 5.25% sodium hypochlorite, citric acid and saline solution. For all of the studied teeth, the step back treating technique was applied, master apical file size 40 and coronal flaring size 70.

The wetness level inside root space was checked by the paper points. After drying, the root canals were filled by an epoxy resin sealer—AH Plus (Dentsply Maillefer, Konstanz, Germany)—and the lateral condensation technique was used. The dental crown was temporarily filled using the self-adhesive posterior restorative system Equia (GC Europe, Leuven, Belgium). The Equia was covered by the apical part of the root to prevent leakage through the apex. The teeth were stored for 7 days in an incubator (at 37 °C, 100% relative humidity). Following this, the palatal cusps of the crowns were removed to the level of the central sulcus, leaving only 2 mm of cingulum above the tooth neck (Figure 1). The palatal canal was prepared using size 3 Pesso drills to obtain a 15 mm length, leaving an apical seal of gutta-percha in the canal (4–6 mm). The teeth were divided into four groups, each group having 15 teeth. Group 1 received prefabricated conventional FRC posts of 1.2 mm diameter (Mirafit White posts, Hager & Werken, Duisburg, Germany). Group 2 received prefabricated conventional FRC posts of 1.2 mm diameter (Rebilda Posts, Voco, Cuxhaven, Germany). Group 3 received one single elastic FRC post of 1.2 mm diameter (EverStick, GC Europe, Leuven, Belgium). Following the manufacturer’s recommendations, the elastic post was inserted into the root canal and adjusted to its shape. After adaptation, the post was removed from the root canal with a needle-nose plier and light-cured for 40 s with an Epilar S10 polymerization lamp (3M Espe, Maplewood, NJ, USA). After polymerization, the EverStick posts became stiff. The cavities in group 4 were restored with a microhybrid composite restorative material, Charisma (Heraeus Kulzer, Hanau, Germany).

The posts in groups 1, 2 and 3 were tried and next their lengths were adjusted. The palatal canal and the prepared tooth surface were etched with orthophosphoric acid for 30 s. The dentin surface was dried with filter paper; next, a bond was applied directly in the canal and on the prepared dentin. The dual-cure self-priming dental adhesive system, Prime and Bond NT (Dentsply, Charlotte, NC, USA), was used for bonding, according to the manufacturer’s instruction. Luting of the posts was performed with a dual resin composite cement, Core.X Flow (Dentsply, Charlotte, NC, USA). In the following step, the crowns of the teeth were reconstructed with Charisma composite material (Heraeus Kulzer, Hanau, Germany). The cellulite core-forming matrices of the same size were used to ensure the uniformity of the specimens. These matrices were fabricated as vacuum-formed foils on a healthy premolar tooth. The core build-up was polymerized for 40 s, then the cellulite forming matrices were removed. Glycerine gel was used, and a final polymerization was performed from each side for 40 s with a Translux Wave polymerization lamp (Kulzer, Milano, Italy). 

After the restorative procedures, the specimens were kept in 0.9% NaCl solution for 1 week. The roots of the prepared teeth were coated with a thin layer of low-density impression material—Oranwash L (Zhermack, Badia Polesine, Italy). It prevented the teeth from rigidly fixing in acrylic blocks and allowed them to maintain their physiological mobility, typical of natural teeth with periodontal fibre (Figure 2). All of the analysed cases were tested in the same way under the same conditions for individual samples. The fixing of the prepared teeth and the method of loading were the same for each case. The teeth were fixed in acrylic blocks into which they were inserted to the level of 2 mm below the tooth neck. The length, width and height of the acrylic blocks were 15 mm for each size. The teeth were fixed at an angle of 90° to the long axis of the tooth in the jaws of a universal testing machine Instron 5944 (Instron, Norwood, MA, USA). A nickel–chrome form reflecting the shape of the chewing surface of a premolar tooth was used as a punch. Acrylic blocks with fixed teeth were loaded in a way to ensure the optimum occlusive contacts of the upper and lower premolar teeth (Figure 3). The loading rate was 1 mm/min. Strength tests were conducted until the tested sample was damaged, i.e., until the first force decrease was recorded in a universal testing machine graph. The maximum failure load was recorded in Newtons (N) (Table 1).

After the examination, the number of damaged teeth that could be rebuilt was assessed. There are two groups of damaged teeth proposed. The first group includes teeth able to be rebuilt. The teeth with the experimental restorable fracture above the cementoenamel junction (CEJ) facture were assigned to the first group. Consequently, the result below the CEJ makes a tooth marked as irreparable and it is likely to be extracted [22].

The distribution of the data from the endurance test of endodontically treated teeth did not differ from normal. The significance of the differences in terms of strength was checked using ANOVA (analysis of variance). The significance level of the hypothesis testing was set at *p* > 0.05. The calculations were conducted in the Statistica version 13.3 program (StatSoft, Kraków, Poland).

### 2.2. Physicochemical Characterization

The crystal structure of the investigated materials was detected using a PANalytical X’Pert Pro X-ray diffractometer (Malvern Panalytical Ltd., Malvern, UK) equipped with Ni-filtered Cu *Kα_1_* radiation (*Kα_1_* = 1.54060 Å, *U* = 40 kV, *I* = 30 mA) in the *2*θ range of 5°–70°. X-ray diffraction (XRD) patterns were parsed by Match! software version 3.7.0.124. Energy-dispersive spectroscopy (EDS) analysis was carried out using an FEI Nova NanoSEM 230 scanning electron microscope (SEM, Hillsboro, OR, USA) equipped with an EDS spectrometer (EDAX GenesisXM4) and operating at an acceleration voltage in the range 3.0–15.0 kV and spots at 2.5–3.0 were observed. The Fourier-transformed infrared spectra were carried out by the Thermo Scientific Nicolet iS50 FT-IR spectrometer (Waltham, MA, USA) equipped with an automated beam splitter exchange system (iS50 ABX containing DLaTGS KBr detector), built-in all-reflective diamond ATR module (iS50 ATR), Thermo Scientific Polaris™ and the HeNe laser was used as an infrared radiation source. Absorption spectra were collected by the Agilent Cary 5000 spectrophotometer (Agilent, 5301 Stevens Creek Blvd, Santa Clara, CA 95051, USA), employing a spectral bandwidth (SBW) with a spectral resolution of 0.25 nm in the visible and ultraviolet range and recorded at room temperature.

## 3. Results

During the experiment, the values of maximum forces, i.e., the forces destroying reconstructed teeth after endodontic treatment, were obtained. The damage could involve dental hard tissues, the post and the light-cured material used in tooth reconstruction. The best results were obtained for teeth reconstructed with Rebilda Posts (mean load 1119 N), while the poorest results were obtained for the teeth in which Mirafit White posts (mean load 968 N) was used (Figure 4). The teeth without any post, reconstructed using only the light-cured material, were damaged by the lowest value of the destruction force (mean 859 N), which confirms the rationale for using posts. Based on the mean values presented in Figure 4, the Rebilda Post group is a more effective treatment than the others. However, in the statistical analysis (*t*-Test ANOVA, *p* > 0.05), no statistically significant differences were found between the various methods. The lack of statistical significance between the investigated treatments may be due to the small number of samples in each group.

After the completion of the strength tests, the assessment of the destructed teeth started. The damage of the supragingival crown, involving natural tissue or composite material, was considered auspicious for further prosthetic reconstruction. Regarding tooth destruction, subgingival damage is considered as the most harmful, especially when it exceeds the biological width. In such cases, obtaining a good impression, removing cement remains or maintaining dryness of the field is required. Any imprecision in the above may lead to periodontitis. Sometimes, even orthodontic extrusion or surgical crown lengthening are necessary to prosthetically restore the lost tooth tissue. Teeth with a broken root are frequently qualified for extraction.

The largest number of supragingival crown fractures in teeth undergoing strength tests were recorded for the samples with Rebilda Posts (60%). It was confirmed that in the case of fibre-reinforced composite posts with good strength parameters, the prognosis for future restoration is better. In the samples with cemented EverStick posts, subgingival fractures were observed more frequently (60%), which made it very difficult or even impossible to prosthetically restore the tooth. The worst prognosis relates to teeth without posts, for 73% of which a subgingival fracture was observed (Table 1 and Table 2).

The presented study investigates three types of fibre-reinforced composite posts made of different chemical compounds. The EverStick post is made of poly (methyl methacrylate) (PMMA) polymer and bisphenol A-glycidyl methacrylate (bis-GMA). The neat and UV light-cured EverStick posts were investigated. The manufacturer of Mirafit White informs us that the material is made of E-glass (a type of glass fibre) and epoxy resin with a 65% fibre content (matrix). In the case of the Rebilda Post, information about the composition includes urethane-dimethacrylate (UDMA) 70 wt.%; glass fibre 20 wt.% and glass filling 10 wt.%.

The X-ray diffraction patterns of the EverStick post, UV light-cured EverStick post and Rebilda Post are presented in Figure 5. In the range of 2 theta from 10° to 25°, two broad diffraction lines that were detected corresponded to PMMA (see Figure 5a). The maximum of the intense one is located at 12° and that of the other one at 22°. The typical amorphous phase is confirmed. The UV light-curing did not affect PMMA crystallization. It could be suggested that glass fibres present in the PMMA matrix precludes the transition of the organized polymeric chain into the crystal phase [23]. The series of diffraction line was found in the XRD pattern of the Rebilda Post. The lines corresponded with the theoretical pattern of orthorhombic phase YbF_3_ (PDF no. 96-153-2779) [24]. In the range of small 2 theta, a high background was detected. It was found to be associated with the amorphous phase of the polymeric matrix. The XRD pattern of the Mirafit White post does not present diffraction lines. 

The presence of the YbF_3_ phase was confirmed by EDS measurements. The existence of Yb^3+^ and F^−^ ions was detected. The results show that the material contains 4.4% of Yb^3+^ and 3.7% of F^−^. Small concentrations of sodium, magnesium, potassium and calcium as well as chlorine ions were also observed. 

Figure 6 shows the FT-IR spectra of the analysed materials and their main chemical components. In general, the FT-IR spectra are composed of a series of lines that are associated with the absorption of the energy by specific chemical bands. Lines are observed in the case of all of the measured samples corresponding to the PMMA molecules. The line at 1716 cm^−1^ confirms the presence of the ester group (C=O). Lines associated with the ether group O–CH_3_ are located at 1163 cm^−1^. The two lines detected at 2963 cm^−1^ and at 2872 cm^−1^ were ascribed to the methyl group (–CH_3_). The methylene group (–CH_2_) was located at 1245 cm^−1^ and at 1434 cm^−1^. The lines at 1510 cm^−1^ were associated with the stretching vibration of an aromatic ring. The stretching vibration of the C–C mode was observed in the range of 1000–800 cm^−1^. In the range of 1484–1389 cm^−1^, absorption lines related to the deformation mode of C–H are detected. The lines observed in the range 1243–1272 cm^−1^ are due to the C–O–C vibration modes. On the spectrum of Mirafit White, the line associated with C–NH_2_ vibration mode was observed at 1507 cm^−1^. It suggests that both the Mirafit White and the Rebilda Post are made of UDMA [25,26,27,28].

The absorbance spectra of UV-vis radiation of UV light-cured the EverStick post were carried out in the range of 250–800 nm as a function of UV light curing time (see Figure 7). Based on the previous papers, it is known that PMMA does not absorb in the range of UV-vis radiation [29]. Absorption bands in the range of 250–330 nm were associated with the presence of bis-GMA [30]. It was found that the relative absorbance value changed with the UV light curing time. The Lambert–Beer law states that the absorption quantity depends on the concentration of a substance, the path length and the proportionality constant. It could be suggested that during the polymerization process, the thickness of the EverStick post has changed. It was calculated that the thickness decreases by 8% after 60 s of UV light curing.

## 4. Discussion

Premolars after endodontic treatment were found to be the most frequently fractured teeth [31,32]. A fracture analysis of maxillary premolars detected that teeth with a damage palatal cusp were the most prone to crack under compressive loading [33]. Therefore, in this study, premolars were prepared for strength tests.

Tooth restoration following endodontic treatment is the main objective of dental prosthetics [34]. It is recommended not to insert posts at the cost of the root dentin [35,36]. The research has shown that excessive preparation for a post not only weakens the tooth structure but may also lead to fractures and defects which could result in increasing the probability of tooth fractures or even tooth loss [37,38]. The use of fibre-reinforced composite posts to reduce the risk of tooth fractures has been questioned many times. Phebus et al. demonstrated that the teeth with a cemented fibre-reinforced composite post were significantly stronger than those which were endodontically treated without the use of a post [39]. They used thermal cycles to test a group of endodontically treated incisor teeth with cemented fibre-reinforced composite posts and another group of endodontically treated teeth without such posts. The thermal cycles simulated the changing conditions in the oral cavity. Lassila et al. discovered that significant influence on the fracture load and flexural strength is related to a thermocycling process for FRC posts. [21]. The flexural modulus decrease was estimated about 10% as a result of the thermocycling process. Moreover, the strength and fracture load decreased by about 18%. Hashemikamangar et al. claimed that no significant effect was detected after the thermocycling process on pushing-out bond strength between the fiber posts and resin core [40]. Moreover, the bond strength values in the thermocycled samples were slightly lower than the values in the nonthermocycled samples.

Based on the statistical analyses, the most homogeneous study group was the teeth rebuilt based on Rebilda Posts. In this group, the highest average value of force needed for destroying the treated tooth was obtained. The control group shows the greatest variability of results, where for the mean value of the destructive force of 859 N, SD is 200. The heterogeneity of the results in this group is caused by the absence of a post and the weakening of the crown resulting from endodontic treatment. The results of the strength tests are satisfactory in terms of the value of the standard deviation for the individual test groups. The heterogeneity of the results shows that the tested samples are biological material. The teeth were obtained from different patients of both sexes and different ages. For this reason, it is not possible to obtain perfectly homogeneous samples of a natural origin. The described studies were of a comparative and pilot nature, aimed at justifying the choice of treatment.

In the authors’ own research, the teeth with cemented EverStick posts showed lower values of strength parameters than the teeth with standard posts. Similar research was conducted by Cagidiaco et al. who observed the survival rate of endodontically treated premolar teeth for three years [41]. The tested groups encompassed: teeth with cemented fibre-reinforced composite posts—D.T. Light-Post (VDW, Munich, Germany), teeth with individually formed posts (EverStick) and endodontically treated premolar teeth without posts. All of the teeth were covered with a metal ceramic crown. In the examined groups, the 36-month survival rate was 76.7%. The smallest percentage of success was recorded for the teeth without posts (62.5%), while in the case of teeth with the cemented prefabricated D.T. Light-Posts, the success rate was higher (90.9%) than in the case of teeth with the cemented EverStick posts (76.7%).

Ferrari et al. also conducted observations of the survival rate of endodontically treated premolar teeth for six years [42]. The research was conducted on 345 patients with teeth restored using fibre-reinforced composite posts (D.T. Light-Post), individually made posts (EverStick) and the control group were premolar teeth restored using only composite material. All of the teeth were covered with a metal ceramic crown. In the case when all of the walls of the treated teeth were preserved, the success percentage for all of the tested groups was the same and was 100%. This means that no complications were recorded in the teeth restored using the standard and individually made posts, and in the group without posts. The results were different in the teeth with a smaller number of preserved walls. In the case when two walls were preserved, the highest success rate was recorded in the group restored with the standard posts (88.9%), followed by the individually made ones (66.7%) and the lowest success rate was reported for the teeth without posts (52.9%). When only one wall was preserved, the success rate was lower and was 77.8% for teeth with the standard posts, 50% for the individually made posts and 29.4% for teeth without posts. These results are close to the authors’ own research results in which the greatest force was needed to destroy crowns in the group of teeth with the standard posts. The higher value of force needed to damage the tooth crown should translate into a higher success rate in treated teeth.

According to the Kivac et al. [43], the presence and the type of the post used did not affect the mechanical strength of the tested teeth, which is a different conclusion from the results obtained in our own research. In the authors’ own research, the ANOVA revealed significant differences (*p* = 0.05) in fracture loads, flexural strengths and flexural modulus of the FRC-post systems tested. The teeth with the cemented EverStick posts showed lower values of strength parameters than the teeth with the standard posts. The teeth without any post, reconstructed using only the light-cured material, were damaged by the lowest value of the destruction force.

Fractures of posts or of restored teeth are among the most common failures. A statistical analysis showed that specimens restored without the use of a post were more frequently used for nonrestorable fractures with the fracture line below the CEJ. Prefabricated posts exhibited more favourable fracture patterns of teeth in comparison with the teeth restored using elastic posts. Thus, the null hypothesis concerning fracture patterns was adopted. It is contrary to the studies of Frater et al. [44] who showed that teeth rebuilt with an elastic FRC post had significantly higher fracture resistance than teeth rebuilt with a prefabricated FRC post. Nevertheless, the load tested by Frater and co-workers was applied at 45° to the long axis of the tooth by aligning a stainless-steel ball-shaped stylus with the occlusal surface. It is a completely different than this investigation.

In our own study, elastic FRC posts showed lower flexural properties than the prefabricated FRC posts [45]. There were statistically significant differences in fracture loads, flexural modulus and flexural strengths during the three-point test. This confirms that posts with a good performance during three-point test offer improved damage resistance after endodontic treatment.

Another study examined the flexural properties of different types of prefabricated FRC posts (Snowpost, Carbopost, Parapost, C-post, Glassix, Carbonite) and compared those values with an individually polymerized FRC EverStick post [19]. The highest flexural strength was obtained for EverStick. It was noted that this effect could be explained by the combination of the polymer matrix and fiber properties building the composite. Furthermore, the presence of PMMA chains in the crosslinked polymer matrix is responsible for the differences between everStick and the other investigated FRC posts. PMMA chains plasticize the crosslinked bis-GMA-based matrix of the EverStick FRC, and thus reduce the generation stresses at the fibre–matrix interface during deflection. Consequently, it could be supposed to improve the strength of everStick FRC material.

## 5. Conclusions

The use of a fibre-reinforced composite post increases the damage resistance in endodontically treated teeth in comparison with teeth restored using light-curing material. Prefabricated Rebilda Posts showed more favourable fracture patterns than the other examined groups.

A statistically significant difference was observed between the use of prefabricated and custom-made FRC posts for multiple restorations. In teeth rebuilt with an elastic FRC post, subgingival fractures were observed more frequently—which made it very difficult or even impossible to prosthetically restore the tooth.

## Figures and Tables

**Figure 1 materials-14-06214-f001:**
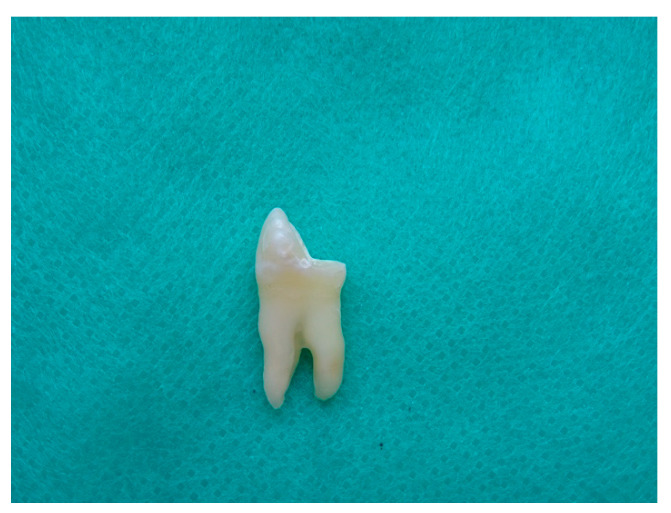
Premolar tooth with removed palatal tubercle.

**Figure 2 materials-14-06214-f002:**
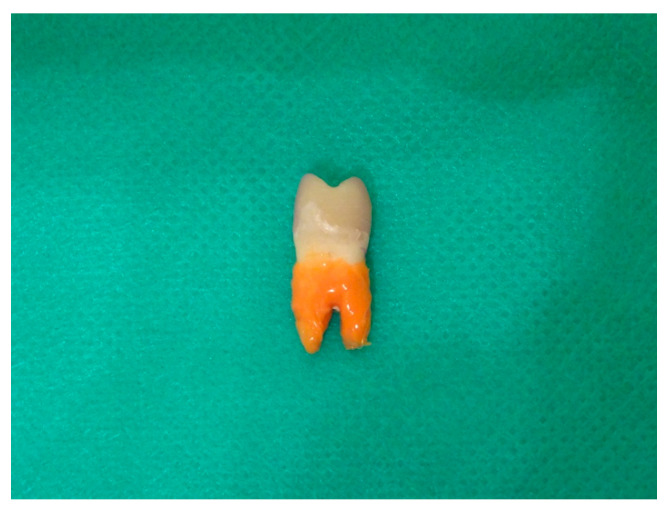
A root covered with a layer of silicone.

**Figure 3 materials-14-06214-f003:**
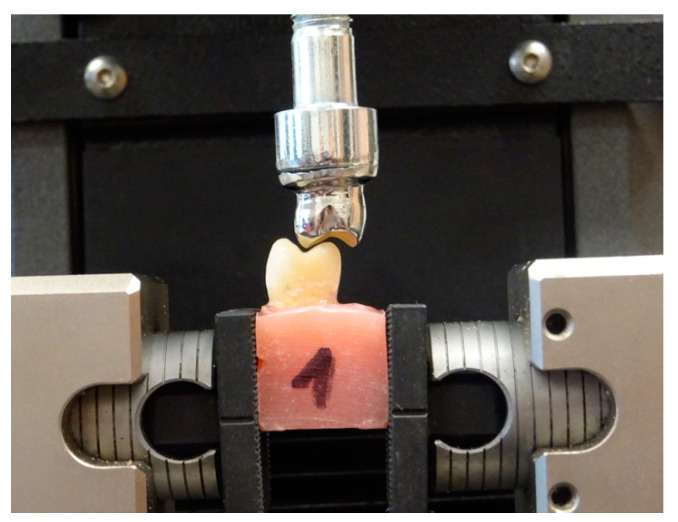
Sample orientation for best occlusal contacts.

**Figure 4 materials-14-06214-f004:**
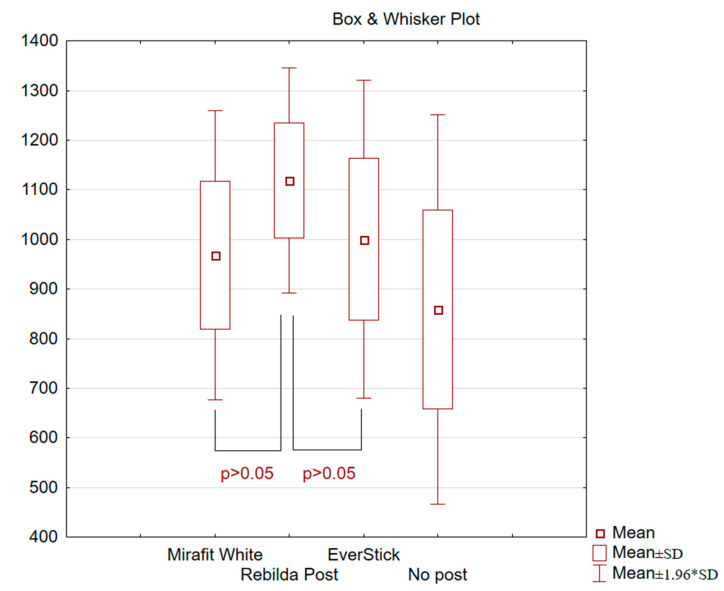
Mean fracture damage values, standard deviations (SD).

**Figure 5 materials-14-06214-f005:**
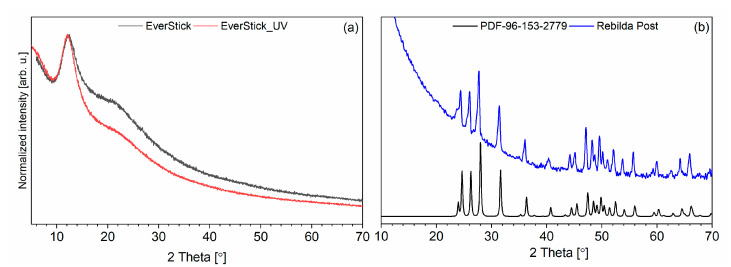
X-ray diffraction pattern of neat EverStick post and UV light-cured EverStick post (**a**) as well as Rebilda Post and PDF theoretical pattern of YbF_3_ (PDF-96-153-2779) (**b**).

**Figure 6 materials-14-06214-f006:**
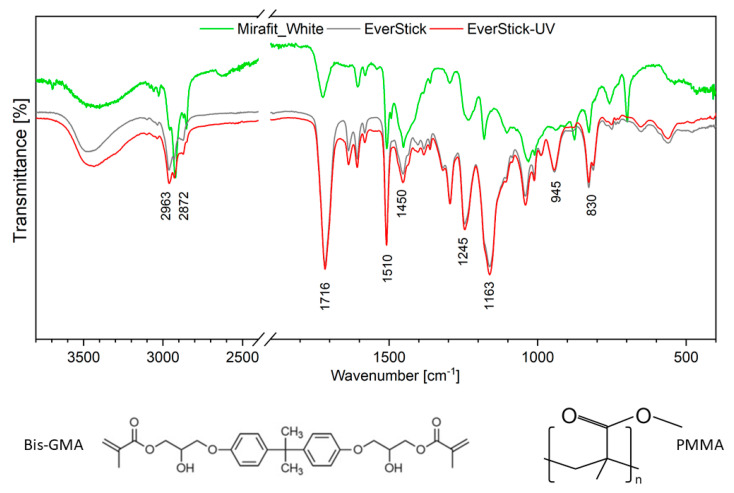
Fourier-transformed mid-infrared spectra of neat EverStick post, UV light-cured EverStick post, Mirafit White and GC Fibre post (**top**) and the structural formula of bisphenol A–glycidyl methacrylate (bis–GMA) (**bottom left**) and poly (methyl methacrylate) PMMA (**bottom right**).

**Figure 7 materials-14-06214-f007:**
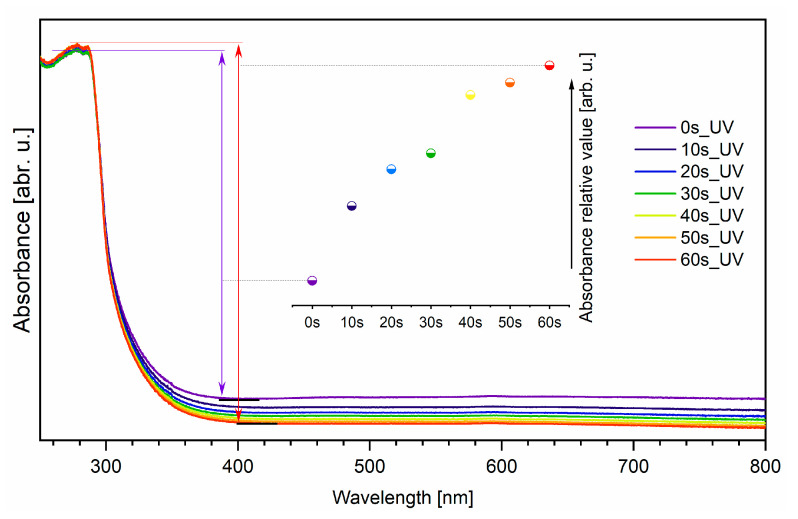
Absorption spectra of neat EverStick post and UV light-cured EverStick post as a function of light-curing time. Inset—The amplitude between the maximum and minimum absorption value.

**Table 1 materials-14-06214-t001:** Fracture thresholds as well as the significance of their difference of studied fibre-reinforced teeth compared to the control group.

Post Type	Valid N	Mean [N]	Min [N]	Max [N]	SD
Mirafit White	15	968	760	1222	149
Rebilda Post	15	1119	973	1323	116
EverStick	15	1000	776	1368	163
No Post	15	859	587	1253	200

**Table 2 materials-14-06214-t002:** Prognosis of tooth restoration possibilities after conducted strength test.

Type of FRC Post	Good Prognosis (Supragingival Fracture)	Poor Prognosis (Subgingival Fracture)
Mirafit White	53% (8/15)	47% (7/15)
Rebilda Post	60% (9/15)	40% (6/15)
EverStick	40% (6/15)	60% (9/15)
No Post	27% (4/15)	73% (11/15)

## Data Availability

Data available on request due to restrictions e.g., privacy or ethical. The data presented in this study are available on request from the corresponding author.

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
