# Peer review of "In Vitro Fracture Resistance of Endodontically Treated Premolar Teeth Restored with Prefabricated and Custom-Made Fibre-Reinforced Composite Posts"

_materials, 2021, doi:10.3390/ma14206214_

Round 1
Reviewer 1 Report
This study was interesting: the methods were adequately described, the results were clearly presented, the conclusions were supported by the results. My suggestions for the authors:
- Could you please include in the methods section the calculation of the minimal sample size of the studied groups?
- Please check the software name( page 4, line 4: Match! soft version...)
- In the discussion section, paragraph second (page 9): I would strongly sugest to rewrite this paragraph answering the question why do the conclusions of Kivac et al. study were different from yours, but not only rewrite the study methods.
Author Response
Dear Editor,
We would like to express our sincerest gratitude to the Reviewer for the enormous efforts in criticizing the manuscript. We have taken into account all raised question here follows the detailed answers to the Reviewer. All changes we have made to the original manuscript, are marked in the red colour in the text.
Reviewer 1
Q1: Could you please include in the methods section the calculation of the minimal sample size of the studied groups?
Response 1: All of the selected teeth had similar dimensions: the length of the crowns was 8,5 mm, the roots’ length was 14 mm, the diameter of the mesial-distal crowns was 7 mm (L75 to L77). The length, width and height of the acrylic blocks were 15mm in each size (L126 to L127).
Q2: Please check the software name (page 4, line 4: Match! soft version...)
Response 2: The software name is correct.
Q 3: In the discussion section, paragraph second (page 9): I would strongly suggest to rewrite this paragraph answering the question why the conclusions of Kivac study do et al. were different from yours, but not only rewrite the study methods.
Response 3: According to the Kivac et al., the presence and the type of the used post did not affect the mechanical strength of the tested teeth, which is a different conclusion from the results obtained in own research. In the authors’ own research, the ANOVA revealed significant differences (p = 0.05) in fracture loads, flexural strengths and flexural modulus of the FRC-post systems tested. The teeth with cemented everStick posts showed lower values of strength parameters than the teeth with standard posts. The teeth without any post, reconstructed using only the light-cured material, were damaged by the lowest value of the destruction force (L317 to L324).
Reviewer 2 Report
The authors aimed to assess the resistance of endodontically treated premolars restored with prefabricated and custom-made fibre-reinforced composite posts, in an in vitro setting. The research question is relevant and the topic is under the scope of the journal. Methods used are adequate and the materials tested are a new clinical option available, which needs further research to provide data to support clinical decision-making process. The manuscript is mostly well-written and the references appropriate and well-cited.
However, there are some concerns which need to be solved before publication:
Abstract:
First use of abbreviatures must be preceded by full description. Absolute values registered for the most relevant groups and statistical significance should be presented in the abstract, at results part.
Introduction:
Authors must acknowledge clinical studies reporting that post space preparation can expose root canal fillings to coronal microleakage and increase the risk of failure. Moreover, exposure of the canals to the autologous microbiota is also a risk factor for development of periapical inflammation https://doi.org/10.1016/j.joen.2013.10.023
Materials and methods:
Change the term palatine to palatal.
P2L86, typo error, received.
Results:
In Figure 4 authors should express the presence or absence of statistical differences between analyzed groups.
L169 to L185, are not results, should be moved to discussion section.
Indicate manufacturer, city and country for all materials and equipment used.
At table 1 and table 2, designations used to identify the groups must be presented in a consistent form.
Discussion:
P8L268 and L272, it is confusing to understand which authors and research projects are being cited. Do authors refer to the present study?
Limitations of the translation of the findings from this in vitro study to the clinical scenario must be pointed, as well as the use of thermocycling, ageing and enzymatic challenge as possible in vitro or ex vivo conditions which may increase the external validity of further studies in this field.
Author Response
Dear Editor,
We would like to express our sincerest gratitude to the Reviewer for the enormous efforts in criticizing the manuscript. We have taken into account all raised question here follows the detailed answers to the Reviewer. All changes we have made to the original manuscript, are marked in the red colour in the text.
Reviewer 2
Q 1: Abstract: First use of abbreviatures must be preceded by full description. Absolute values registered for the most relevant groups and statistical significance should be presented in the abstract, at results part.
Response 1: We would like to thank for the suggestion. The description was corrected, and the changes were matched with the red colour in the main text: L13; L24-L26.
Q 2: Introduction: Authors must acknowledge clinical studies reporting that post space preparation can expose root canal fillings to coronal microleakage and increase the risk of failure. Moreover, exposure of the canals to the autologous microbiota is also a risk factor for development of periapical inflammation https://doi.org/10.1016/j.joen.2013.10.023
Response 2: During mechanical preparation for the post space, it is possible that the root filling may be twisted or vibrated, with disruption of the seal [R1]. Obturated root canals may be recontaminated by micro-organisms during preparation of post space for the provision of a post- retained restoration when the remaining apical section of the root filling is of inadequte density or length. It is a risk factor for development of periapical inflammation. Santos et all assess periapical inflammation subsequent to coronal inoculation of dog teeth root [R2]. They concluded that roots treated in 2 sessions had over 2-fold more periapical inflammation than roots treated in 1 session 7months after treatment.
Q 3: Materials and methods: Change the term palatine to palatal. P2L86, typo error, received.
Response 3: It was changed and marked with red text. Please see line L94.
Q 4: Results:
In Figure 4 authors should express the presence or absence of statistical differences between analyzed groups.
L169 to L185, are not results, should be moved to discussion section.
Indicate manufacturer, city and country for all materials and equipment used.
At table 1 and table 2, designations used to identify the groups must be presented in a consistent form.
Response 4: The suggestion about Figure 4 is not understandable. In the Figure 4 was shown the value of the force needed to destroy the investigated tooth. The differences were presented as symbol with the error bars.
L169 to L185 moved to discussion section L276 to L287.
The manufacturer, city and country for all materials and equipment were added.
Table 1 and table 2 were improved.
Q5: Discussion:
P8L268 and L272, it is confusing to understand which authors and research projects are being cited. Do authors refer to the present study?
Limitations of the translation of the findings from this in vitro study to the clinical scenario must be pointed, as well as the use of thermocycling, ageing and enzymatic challenge as possible in vitro or ex vivo conditions which may increase the external validity of further studies in this field.
Response 5: Phebus et al. demonstrated that the teeth with a cemented fibre-reinforced composite post were significantly stronger than those which were endodontically treated without the use of a post. They used thermal cycles to test a group of endodontically treated incisor teeth with cemented fibre-reinforced composite posts and another group of endodontically treated teeth without such posts. The thermal cycles simulated changing conditions in the oral cavity. Lassila et al. revealed that thermocycling for FRC posts had a significant effect on the fracture load and flexural strength. In general, thermocycling decreased the flexural modulus of the tested FRC posts by about 10%. Strength and fracture load decreased by about 18%. Hashemikamangar et al. claimed that thermocycling had no significant effect on push-out bond strength of fiber posts to resin core [R3]. Moreover, the bond strength values in the thermocycled samples were slightly lower than the values in the non-thermocycled samples (L263 to L275).
References
[R1] Saunders WP, Saunders EM. Coronal leakage as a cause of failure in root-canal therapy: a review. Endod Dent Traumatol 1994; 10:105–8.
[R2] Periapical Inflammation Subsequent to Coronal Inoculation of Dog Teeth Root Filled with Resilon/Epiphany in 1 or 2 Treatment Sessions with Chlorhexidine Medication João M. Santos Paulo J. Palma, João C. Ramos, António S. Cabrita, Shimon Friedman, Published:December 16, 2013DOI:https://doi.org/10.1016/j.joen.2013.10.023 (L50 to L57).
[R3] Hashemikamangar SS, Hasanitabatabaee M, Kalantari S, Gholampourdehaky M, Ranjbaromrani L, Ebrahimi H. Bond strength of fiber posts to composite core: Effect of surface treatment with Er,Cr:YSGG laser and thermocycling. J Lasers Med Sci. 2018;9(1):36–42.
Round 2
Reviewer 2 Report
The authors performed some modifications the quality of the manuscript improved from the previous version. However, there are still some details to be addressed:
- The important finding from the suggested study https://doi.org/10.1016/j.joen.2013.10.023
is the fact that root canal filled teeth coronaly restored with rubber dam did not develop severe periapical inflammation but root canal fillings exposed to microbial challenge with exposure to saliva developed severe periapical inflammation, which shows that current root canal filling methods only partially inhibited coronal microbial ingress and the development of post-treatment apical periodontitis, highlighting the risk of periapical pathology in case of break in chain of asepsis during post preparation. The text must be modified accordingly.
- Results: I am sorry but in the version of the manuscript that I received for second review, I still do not see in Figure 4 indication of presence or absence of statistical differences between groups presented in the graphic.
Author Response
Dear Editor,
We would like to express our sincerest gratitude once again to the Reviewer for the enormous efforts in criticizing the manuscript. We have taken into account all raised question here follows the detailed answers to the Reviewer. All changes we have made to the original manuscript, are marked in the red colour in the text.
Reviewer 2
Q1: The important finding from the suggested study https://doi.org/10.1016/j.joen.2013.10.023 is the fact that root canal filled teeth coronally restored with rubber dam did not develop severe periapical inflammation but root canal fillings exposed to microbial challenge with exposure to saliva developed severe periapical inflammation, which shows that current root canal filling methods only partially inhibited coronal microbial ingress and the development of post-treatment apical periodontitis, highlighting the risk of periapical pathology in case of break in chain of asepsis during post preparation. The text must be modified accordingly.
Response 1: We are grateful for your suggestion. The main text was modified, see lines 55-60. All changes were marked with red colour.
Q2: Results: I am sorry but in the version of the manuscript that I received for second review, I still do not see in Figure 4 indication of presence or absence of statistical differences between groups presented in the graphic.
Response 2: As can be seen that the Rebilda Post group is a more effective treatment than the others and it is a basis for the mean values presented in Fig. 4. However, in the statistical analysis (T-Test ANOVA, p> 0.05), no statistically significant difference was found between the various methods. The lack of statistical significance between the investigated treatments may be due to the small number of samples in each group. The main text was improved (see lines 177-181) and the Figure 4 was changed.
